# Evaluating Robustness of Neural Networks with Mixed Integer Programming

**Vincent Tjeng, Kai Xiao, Russ Tedrake**
Massachusetts Institute of Technology
{vtjeng, kaix, russt}@mit.edu

## Abstract

Neural networks trained only to optimize for training accuracy can often be fooled by adversarial examples — slightly perturbed inputs misclassified with high confidence. Verification of networks enables us to gauge their vulnerability to such adversarial examples. We formulate verification of piecewise-linear neural networks as a mixed integer program. On a representative task of finding minimum adversarial distortions, our verifier is two to three orders of magnitude quicker than the state-of-the-art. We achieve this computational speedup via tight formulations for non-linearities, as well as a novel presolve algorithm that makes full use of all information available. The computational speedup allows us to verify properties on convolutional and residual networks with over 100,000 ReLUs — several orders of magnitude more than networks previously verified by any complete verifier. In particular, we determine for the first time the *exact* adversarial accuracy of an MNIST classifier to perturbations with bounded $l_\infty$ norm $\epsilon = 0.1$: for this classifier, we find an adversarial example for 4.38% of samples, and a certificate of robustness to norm-bounded perturbations for the remainder. Across all robust training procedures and network architectures considered, and for both the MNIST and CIFAR-10 datasets, we are able to certify more samples than the state-of-the-art *and* find more adversarial examples than a strong first-order attack.

## 1 Introduction

Neural networks trained only to optimize for training accuracy have been shown to be vulnerable to *adversarial examples*: perturbed inputs that are very similar to some regular input but for which the output is radically different (Szegedy et al., 2014). There is now a large body of work proposing defense methods to produce classifiers that are more robust to adversarial examples. However, as long as a defense is evaluated only via heuristic attacks (such as the Fast Gradient Sign Method (FGSM) (Goodfellow et al., 2015) or Carlini & Wagner (2017b)'s attack (CW)), we have no guarantee that the defense actually increases the robustness of the classifier produced. Defense methods thought to be successful when published have often later been found to be vulnerable to a new class of attacks. For instance, multiple defense methods are defeated in Carlini & Wagner (2017a) by constructing defense-specific loss functions and in Athalye et al. (2018) by overcoming obfuscated gradients.

Fortunately, we *can* evaluate robustness to adversarial examples in a principled fashion. One option is to determine (for each test input) the minimum distance to the closest adversarial example, which we call the *minimum adversarial distortion* (Carlini et al., 2017). Alternatively, we can determine the *adversarial test accuracy* (Bastani et al., 2016), which is the proportion of the test set for which no perturbation in some bounded class causes a misclassification. An increase in the mean minimum adversarial distortion or in the adversarial test accuracy indicates an improvement in robustness.[1]

We present an efficient implementation of a mixed-integer linear programming (MILP) verifier for properties of piecewise-linear feed-forward neural networks. Our tight formulation for non-linearities and our novel presolve algorithm combine to minimize the number of binary variables in the MILP problem and dramatically improve its numerical conditioning. Optimizations in our MILP

---

[1] The two measures are related: a solver that can find certificates for bounded perturbations can be used iteratively (in a binary search process) to find minimum distortions.

implementation improve performance by several orders of magnitude when compared to a naïve MILP implementation, and we are two to three orders of magnitude faster than the state-of-the-art Satisfiability Modulo Theories (SMT) based verifier, Reluplex (Katz et al., 2017)

We make the following key contributions:

- We demonstrate that, despite considering the full combinatorial nature of the network, our verifier *can* succeed at evaluating the robustness of larger neural networks, including those with convolutional and residual layers.

- We identify *why* we can succeed on larger neural networks with hundreds of thousands of units. First, a large fraction of the ReLUs can be shown to be either always active or always inactive over the bounded input domain. Second, since the predicted label is determined by the unit in the final layer with the maximum activation, proving that a unit *never* has the maximum activation over all bounded perturbations eliminates it from consideration. We exploit both phenomena, reducing the overall number of non-linearities considered.

- We determine for the first time the exact adversarial accuracy for MNIST classifiers to perturbations with bounded $l_\infty$ norm $\epsilon$. We are also able to certify more samples than the state-of-the-art *and* find more adversarial examples across MNIST and CIFAR-10 classifiers with different architectures trained with a variety of robust training procedures.

Our code is available at `https://github.com/vtjeng/MIPVerify.jl`.

## 2 RELATED WORK

Our work relates most closely to other work on verification of piecewise-linear neural networks; Bunel et al. (2018) provides a good overview of the field. We categorize verification procedures as *complete* or *incomplete*. To understand the difference between these two types of procedures, we consider the example of evaluating adversarial accuracy.

As in Kolter & Wong (2017), we call the exact set of all final-layer activations that can be achieved by applying a bounded perturbation to the input the *adversarial polytope*. Incomplete verifiers reason over an *outer approximation* of the adversarial polytope. As a result, when using incomplete verifiers, the answer to some queries about the adversarial polytope may not be decidable. In particular, incomplete verifiers can only certify robustness for a fraction of robust input; the status for the remaining input is undetermined. In contrast, complete verifiers reason over the *exact* adversarial polytope. Given sufficient time, a complete verifier can provide a definite answer to any query about the adversarial polytope. In the context of adversarial accuracy, complete verifiers will obtain a valid adversarial example or a certificate of robustness for *every* input. When a time limit is set, complete verifiers behave like incomplete verifiers, and resolve only a fraction of queries. However, complete verifiers do allow users to answer a larger fraction of queries by extending the set time limit.

Incomplete verifiers for evaluating network robustness employ a range of techniques, including duality (Dvijotham et al., 2018; Kolter & Wong, 2017; Raghunathan et al., 2018), layer-by-layer approximations of the adversarial polytope (Xiang et al., 2018), discretizing the search space (Huang et al., 2017), abstract interpretation (Gehr et al., 2018), bounding the local Lipschitz constant (Weng et al., 2018), or bounding the activation of the ReLU with linear functions (Weng et al., 2018).

Complete verifiers typically employ either MILP solvers as we do (Cheng et al., 2017; Dutta et al., 2018; Fischetti & Jo, 2018; Lomuscio & Maganti, 2017) or SMT solvers (Carlini et al., 2017; Ehlers, 2017; Katz et al., 2017; Scheibler et al., 2015). Our approach improves upon existing MILP-based approaches with a tighter formulation for non-linearities and a novel presolve algorithm that makes full use of all information available, leading to solve times several orders of magnitude faster than a naïvely implemented MILP-based approach. When comparing our approach to the state-of-the-art SMT-based approach (Reluplex) on the task of finding minimum adversarial distortions, we find that our verifier is two to three orders of magnitude faster. Crucially, these improvements in performance allow our verifier to verify a network with over 100,000 units — several orders of magnitude larger than the largest MNIST classifier previously verified with a complete verifier.

A complementary line of research to verification is in robust training procedures that train networks *designed* to be robust to bounded perturbations. Robust training aims to minimize the "worst-case

loss" for each example — that is, the maximum loss over all bounded perturbations of that example (Kolter & Wong, 2017). Since calculating the exact worst-case loss can be computationally costly, robust training procedures typically minimize an estimate of the worst-case loss: either a *lower bound* as is the case for adversarial training (Goodfellow et al., 2015), or an *upper bound* as is the case for certified training approaches (Hein & Andriushchenko, 2017; Kolter & Wong, 2017; Raghunathan et al., 2018). Complete verifiers such as ours can augment robust training procedures by resolving the status of input for which heuristic attacks cannot find an adversarial example and incomplete verifiers cannot guarantee robustness, enabling more accurate comparisons between different training procedures.

## 3 BACKGROUND AND NOTATION

We denote a neural network by a function $f(\cdot; \theta) : \mathbb{R}^m \to \mathbb{R}^n$ parameterized by a (fixed) vector of weights $\theta$. For a classifier, the output layer has a neuron for each target class.

**Verification as solving an MILP**. The general problem of verification is to determine whether some property $P$ on the output of a neural network holds for all input in a bounded input domain $\mathcal{C} \subseteq \mathbb{R}^m$. For the verification problem to be expressible as solving an MILP, $P$ must be expressible as the conjunction or disjunction of linear properties $P_{i,j}$ over some set of polyhedra $\mathcal{C}_i$, where $\mathcal{C} = \cup \mathcal{C}_i$.

In addition, $f(\cdot)$ must be composed of piecewise-linear layers. This is not a particularly restrictive requirement: piecewise-linear layers include linear transformations (such as fully-connected, convolution, and average-pooling layers) and layers that use piecewise-linear functions (such as ReLU or maximum-pooling layers). We provide details on how to express these piecewise-linear functions in the MILP framework in Section 4.1. The "shortcut connections" used in architectures such as ResNet (He et al., 2016) are also linear, and batch normalization (Ioffe & Szegedy, 2015) or dropout (Srivastava et al., 2014) are linear transformations at *evaluation time* (Bunel et al., 2018).

## 4 FORMULATING ROBUSTNESS EVALUATION OF CLASSIFIERS AS AN MILP

**Evaluating Adversarial Accuracy.** Let $\mathcal{G}(x)$ denote the region in the input domain corresponding to all allowable perturbations of a particular input $x$. In general, perturbed inputs must also remain in the domain of valid inputs $\mathcal{X}_{valid}$. For example, for normalized images with pixel values ranging from 0 to 1, $\mathcal{X}_{valid} = [0, 1]^m$. As in Madry et al. (2018), we say that a neural network is robust to perturbations on $x$ if the predicted probability of the true label $\lambda(x)$ exceeds that of every other label for all perturbations:

$$\forall x' \in (\mathcal{G}(x) \cap \mathcal{X}_{valid}) : \ \text{argmax}_i(f_i(x')) = \lambda(x) \tag{1}$$

Equivalently, the network is robust to perturbations on $x$ if and only if Equation 2 is infeasible for $x'$.

$$(x' \in (\mathcal{G}(x) \cap \mathcal{X}_{valid})) \wedge \left( f_{\lambda(x)}(x') < \max_{\mu \in [1,n] \setminus \{\lambda(x)\}} f_\mu(x') \right) \tag{2}$$

where $f_i(\cdot)$ is the $i^{\text{th}}$ output of the network. For conciseness, we call $x$ *robust* with respect to the network if $f(\cdot)$ is robust to perturbations on $x$. If $x$ is not robust, we call any $x'$ satisfying the constraints a *valid* adversarial example to $x$. The *adversarial accuracy* of a network is the fraction of the test set that is robust; the *adversarial error* is the complement of the adversarial accuracy.

As long as $\mathcal{G}(x) \cap \mathcal{X}_{valid}$ can be expressed as the union of a set of polyhedra, the feasibility problem can be expressed as an MILP. The four robust training procedures we consider (Kolter & Wong, 2017; Wong et al., 2018; Madry et al., 2018; Raghunathan et al., 2018) are designed to be robust to perturbations with bounded $l_\infty$ norm, and the $l_\infty$-ball of radius $\epsilon$ around each input $x$ can be succinctly represented by the set of linear constraints $\mathcal{G}(x) = \{x' \mid \forall i : -\epsilon \leq (x - x')_i \leq \epsilon\}$.

**Evaluating Mean Minimum Adversarial Distortion.** Let $d(\cdot, \cdot)$ denote a distance metric that measures the perceptual similarity between two input images. The minimum adversarial distortion under $d$ for input $x$ with true label $\lambda(x)$ corresponds to the solution to the optimization:

$$\min_{x'} d(x', x) \tag{3}$$

$$\text{subject to} \quad \text{argmax}_i(f_i(x')) \neq \lambda(x) \tag{4}$$

$$x' \in \mathcal{X}_{valid} \tag{5}$$

We can *target* the attack to generate an adversarial example that is classified in one of a set of target labels $T$ by replacing Equation 4 with $\text{argmax}_i(f_i(x')) \in T$.

The most prevalent distance metrics in the literature for generating adversarial examples are the $l_1$ (Carlini & Wagner, 2017b; Chen et al., 2018), $l_2$ (Szegedy et al., 2014), and $l_\infty$ (Goodfellow et al., 2015; Papernot et al., 2016) norms. All three can be expressed in the objective without adding any additional integer variables to the model (Boyd & Vandenberghe, 2004); details are in Appendix A.3.

## 4.1 Formulating Piecewise-linear Functions in the MILP framework

Tight formulations of the ReLU and maximum functions are critical to good performance of the MILP solver; we thus present these formulations in detail with accompanying proofs.[2]

**Formulating ReLU**     Let $y = \max(x, 0)$, and $l \leq x \leq u$. There are three possibilities for the *phase* of the ReLU. If $u \leq 0$, we have $y \equiv 0$. We say that such a unit is *stably inactive*. Similarly, if $l \geq 0$, we have $y \equiv x$. We say that such a unit is *stably active*. Otherwise, the unit is *unstable*. For unstable units, we introduce an indicator decision variable $a = \mathbb{1}_{x \geq 0}$. As we prove in Appendix A.1, $y = \max(x, 0)$ is equivalent to the set of linear and integer constraints in Equation 6.[3]

$$(y \leq x - l(1 - a)) \wedge (y \geq x) \wedge (y \leq u \cdot a) \wedge (y \geq 0) \wedge (a \in \{0, 1\}) \tag{6}$$

**Formulating the Maximum Function**     Let $y = \max(x_1, x_2, \ldots, x_m)$, and $l_i \leq x_i \leq u_i$.

*Proposition* 1. Let $l_{max} \triangleq \max(l_1, l_2, \ldots, l_m)$. We can eliminate from consideration all $x_i$ where $u_i \leq l_{max}$, since we know that $y \geq l_{max} \geq u_i \geq x_i$.

We introduce an indicator decision variable $a_i$ for each of our input variables, where $a_i = 1 \implies y = x_i$. Furthermore, we define $u_{max,-i} \triangleq \max_{j \neq i}(u_j)$. As we prove in Appendix A.2, the constraint $y = \max(x_1, x_2, \ldots, x_m)$ is equivalent to the set of linear and integer constraints in Equation 7.

$$\bigwedge_{i=1}^{m} ((y \leq x_i + (1 - a_i)(u_{max,-i} - l_i)) \wedge (y \geq x_i)) \wedge \left(\sum_{i=1}^{m} a_i = 1\right) \wedge (a_i \in \{0, 1\}) \tag{7}$$

## 4.2 Progressive Bounds Tightening

We previously assumed that we had some element-wise bounds on the inputs to non-linearities. In practice, we have to carry out a presolve step to determine these bounds. Determining tight bounds is critical for problem tractability: tight bounds strengthen the problem formulation and thus improve solve times (Vielma, 2015). For instance, if we can prove that the phase of a ReLU is stable, we can avoid introducing a binary variable. More generally, loose bounds on input to some unit will propagate downstream, leading to units in later layers having looser bounds.

We used two procedures to determine bounds: Interval Arithmetic (ia), also used in Cheng et al. (2017); Dutta et al. (2018), and the slower but tighter Linear Programming (lp) approach. Implementation details are in Appendix B.

Since faster procedures achieve efficiency by compromising on tightness of bounds, we face a trade-off between higher *build times* (to determine tighter bounds to inputs to non-linearities), and higher *solve times* (to solve the main MILP problem in Equation 2 or Equation 3-5). While a degree of compromise is inevitable, our knowledge of the non-linearities used in our network allows us to reduce average build times without affecting the strength of the problem formulation.

The key observation is that, for piecewise-linear non-linearities, there are thresholds beyond which further refining a bound will not improve the problem formulation. With this in mind, we adopt a progressive bounds tightening approach: we begin by determining coarse bounds using fast procedures and only spend time refining bounds using procedures with higher computational complexity if doing so could provide additional information to improve the problem formulation.[4] Pseudocode

---

[2]Huchette & Vielma (2017) presents formulations for general piecewise linear functions.

[3]We note that relaxing the binary constraint $a \in \{0, 1\}$ to $0 \leq a \leq 1$ results in the convex formulation for the ReLU in Ehlers (2017)

[4]As a corollary, we always use only ia for the output of the first layer, since the independence of network input implies that ia is provably optimal for that layer.

demonstrating how to efficiently determine bounds for the tightest possible formulations for the ReLU and maximum function is provided below and in Appendix C respectively.

GETBOUNDSFORRELU($x, fs$)

1   ▷ $fs$ are the procedures to determine bounds, sorted in increasing computational complexity.
2   $l_{best} = -\infty$; $u_{best} = \infty$   ▷ initialize best known upper and lower bounds on $x$
3   **for** $f$ in $fs$:   ▷ carrying out progressive bounds tightening
4      **do** $u = f(x, boundType = upper)$; $u_{best} = \min(u_{best}, u)$
5      **if** $u_{best} \leq 0$ **return** $(l_{best}, u_{best})$   ▷ Early return: $x \leq u_{best} \leq 0$; thus $\max(x, 0) \equiv 0$.
6      $l = f(x, boundType = lower)$; $l_{best} = \max(l_{best}, l)$
7      **if** $l_{best} \geq 0$ **return** $(l_{best}, u_{best})$   ▷ Early return: $x \geq l_{best} \geq 0$; thus $\max(x, 0) \equiv x$
8   **return** $(l_{best}, u_{best})$   ▷ $x$ could be either positive or negative.

The process of progressive bounds tightening is naturally extensible to more procedures. Kolter & Wong (2017); Wong et al. (2018); Dvijotham et al. (2018); Weng et al. (2018) each discuss procedures to determine bounds with computational complexity and tightness intermediate between IA and LP. Using one of these procedures in addition to IA and LP has the potential to further reduce build times.

# 5 EXPERIMENTS

**Dataset.** All experiments are carried out on classifiers for the MNIST dataset of handwritten digits or the CIFAR-10 dataset of color images.

**Architectures.** We conduct experiments on a range of feed-forward networks. In all cases, ReLUs follow each layer except the output layer. MLP-$m \times [n]$ refers to a multilayer perceptron with $m$ hidden layers and $n$ units per hidden layer. We further abbreviate MLP-$1 \times [500]$ and MLP-$2 \times [200]$ as **MLP$_A$** and **MLP$_B$** respectively. **CNN$_A$** and **CNN$_B$** refer to the *small* and *large* ConvNet architectures in Wong et al. (2018). CNN$_A$ has two convolutional layers (stride length 2) with 16 and 32 filters (size $4 \times 4$) respectively, followed by a fully-connected layer with 100 units. CNN$_B$ has four convolutional layers with 32, 32, 64, and 64 filters, followed by two fully-connected layers with 512 units. **RES** refers to the ResNet architecture used in Wong et al. (2018), with 9 convolutional layers in four blocks, followed by two fully-connected layers with 4096 and 1000 units respectively.

**Training Methods.** We conduct experiments on networks trained with a regular loss function and networks trained to be robust. Networks trained to be robust are identified by a prefix corresponding to the method used to approximate the worst-case loss: **LP$_d$**[5] when the dual of a linear program is used, as in Kolter & Wong (2017); **SDP$_d$** when the dual of a semidefinite relaxation is used, as in Raghunathan et al. (2018); and **Adv** when adversarial examples generated via Projected Gradient Descent (PGD) are used, as in Madry et al. (2018). Full details on each network are in Appendix D.1.

**Experimental Setup.** We run experiments on a modest 8 CPUs@2.20 GHz with 8GB of RAM. Appendix D.2 provides additional details about the computational environment. Maximum build effort is LP. Unless otherwise noted, we report a timeout if solve time for some input exceeds 1200s.

## 5.1 PERFORMANCE COMPARISONS

### 5.1.1 COMPARISONS TO OTHER MILP-BASED COMPLETE VERIFIERS

Our MILP approach implements three key optimizations: we use *progressive tightening*, make use of the information provided by the *restricted input domain* $\mathcal{G}(x)$, and use *asymmetric bounds* in the ReLU formulation in Equation 6. None of the four other MILP-based complete verifiers implement *progressive tightening* or use the *restricted input domain*, and only Fischetti & Jo (2018) uses *asymmetric bounds*. Since none of the four verifiers have publicly available code, we use ablation tests to provide an idea of the difference in performance between our verifier and these existing ones.

When removing *progressive tightening*, we directly use LP rather than doing IA first. When removing *using restricted input domain*, we determine bounds under the assumption that our perturbed input could be anywhere in the full input domain $\mathcal{X}_{valid}$, imposing the constraint $x' \in \mathcal{G}(x)$ only after all

---

[5]This is unrelated to the procedure to determine bounds named LP.

bounds are determined. Finally, when removing *using asymmetric bounds*, we replace $l$ and $u$ in Equation 6 with $-M$ and $M$ respectively, where $M \triangleq \max(-l, u)$, as is done in Cheng et al. (2017); Dutta et al. (2018); Lomuscio & Maganti (2017). We carry out experiments on an MNIST classifier; results are reported in Table 1.

Table 1: Results of ablation testing on our verifier, where each test removes a single optimization. The task was to determine the adversarial accuracy of the MNIST classifier $LP_d$-$CNN_A$ to perturbations with $l_\infty$ norm-bound $\epsilon = 0.1$. *Build* time refers to time used to determine bounds, while *solve* time refers to time used to solve the main MILP problem in Equation 2 once all bounds have been determined. During solve time, we solve a linear program for each of the *nodes explored* in the MILP search tree.
[†]We exclude the initial build time required (3593s) to determine reusable bounds.

| Optimization Removed | Mean Time / s | | | Nodes Explored | | Fraction Timed Out |
|---|---|---|---|---|---|---|
| | Build | Solve | Total | Mean | Median | |
| *(Control)* | 3.44 | 0.08 | 3.52 | 1.91 | 0 | 0 |
| Progressive tightening | 7.66 | 0.11 | 7.77 | 1.91 | 0 | 0 |
| Using restricted input domain[†] | 1.49 | 56.47 | 57.96 | 649.63 | 65 | 0.0047 |
| Using asymmetric bounds | 4465.11 | 133.03 | 4598.15 | 1279.06 | 105 | 0.0300 |

The ablation tests demonstrate that each optimization is critical to the performance of our verifier. In terms of performance comparisons, we expect our verifier to have a runtime several orders of magnitude faster than any of the three verifiers not using *asymmetric bounds*. While Fischetti & Jo (2018) do use *asymmetric bounds*, they do not use information from the *restricted input domain*; we thus expect our verifier to have a runtime at least an order of magnitude faster than theirs.

### 5.1.2 COMPARISONS TO OTHER COMPLETE AND INCOMPLETE VERIFIERS

We also compared our verifier to other verifiers on the task of finding minimum targeted adversarial distortions for MNIST test samples. Verifiers included for comparison are 1) `Reluplex` (Katz et al., 2017), a complete verifier also able to find the true minimum distortion; and 2) `LP`[6], `Fast-Lip`, `Fast-Lin` (Weng et al., 2018), and `LP-full` (Kolter & Wong, 2017), incomplete verifiers that provide a certified *lower* bound on the minimum distortion.

**Verification Times, vis-à-vis the state-of-the-art SMT-based complete verifier `Reluplex`.** Figure 1 presents average verification times per sample. All solves for our method were run to completion. On the $l_\infty$ norm, we improve on the speed of `Reluplex` by two to three orders of magnitude.

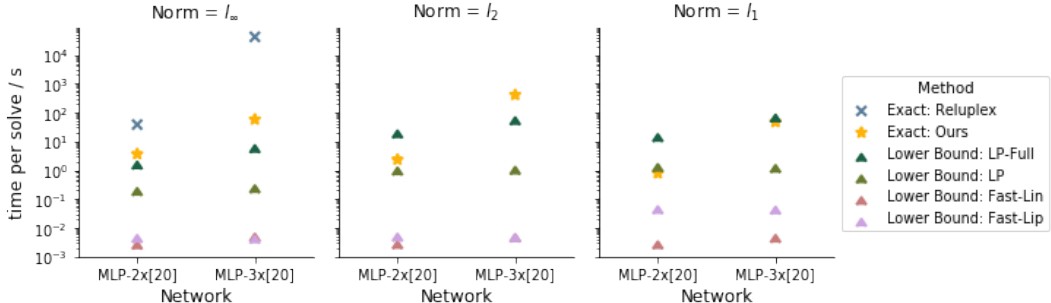

Figure 1: Average times for determining bounds on or exact values of minimum targeted adversarial distortion for MNIST test samples. We improve on the speed of the state-of-the-art complete verifier `Reluplex` by two to three orders of magnitude. Results for methods other than ours are from Weng et al. (2018); results for `Reluplex` were only available in Weng et al. (2018) for the $l_\infty$ norm.

**Minimum Targeted Adversarial Distortions, vis-à-vis incomplete verifiers.** Figure 2 compares lower bounds from the incomplete verifiers to the exact value we obtain. The gap between the best

---

[6]This is unrelated to the procedure to determine bounds named LP, or the training procedure $LP_d$.

lower bound and the true minimum adversarial distortion is significant even on these small networks. This corroborates the observation in Raghunathan et al. (2018) that incomplete verifiers provide weak bounds if the network they are applied to is not optimized for that verifier. For example, under the $l_\infty$ norm, the best certified lower bound is less than half of the true minimum distortion. In context: a network robust to perturbations with $l_\infty$ norm-bound $\epsilon = 0.1$ would only be verifiable to $\epsilon = 0.05$.

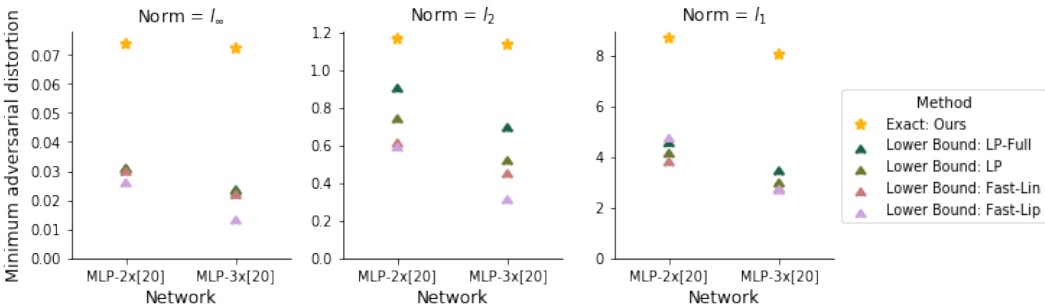

Figure 2: Bounds on or exact values of minimum targeted adversarial distortion for MNIST test samples. The gap between the true minimum adversarial distortion and the best lower bound is significant in all cases, increasing for deeper networks. We report mean values over 100 samples.

## 5.2 DETERMINING ADVERSARIAL ACCURACY OF MNIST AND CIFAR-10 CLASSIFIERS

We use our verifier to determine the adversarial accuracy of classifiers trained by a range of robust training procedures on the MNIST and CIFAR-10 datasets. Table 2 presents the test error and estimates of the adversarial error for these classifiers.[7] For MNIST, we verified a range of networks trained to be robust to attacks with bounded $l_\infty$ norm $\epsilon = 0.1$, as well as networks trained to be robust to larger attacks of $\epsilon = 0.2, 0.3$ and $0.4$. Lower bounds on the adversarial error are proven by providing adversarial examples for input that is not robust. We compare the number of samples for which we successfully find adversarial examples to the number for PGD, a strong first-order attack. Upper bounds on the adversarial error are proven by providing certificates of robustness for input that is robust. We compare our upper bounds to the previous state-of-the-art for each network.

While performance depends on the training method and architecture, we improve on both the lower and upper bounds for *every* network tested.[8] For lower bounds, we successfully find an adversarial example for every test sample that PGD finds an adversarial example for. In addition, we observe that PGD 'misses' some valid adversarial examples: it fails to find these adversarial examples even though they are within the norm bounds. As the last three rows of Table 2 show, PGD misses for a larger fraction of test samples when $\epsilon$ is larger. We also found that PGD is far more likely to miss for some test sample if the minimum adversarial distortion for that sample is close to $\epsilon$; this observation is discussed in more depth in Appendix G. For upper bounds, we improve on the bound on adversarial error even when the upper bound on the worst-case loss — which is used to generate the certificate of robustness — is *explicitly* optimized for during training (as is the case for LP$_d$ and SDP$_d$ training). Our method also scales well to the more complex CIFAR-10 dataset and the larger LP$_d$-RES network (which has 107,496 units), with the solver reaching the time limit for only 0.31% of samples.

Most importantly, we are able to determine the **exact adversarial accuracy** for Adv-MLP$_B$ and LP$_d$-CNN$_A$ for all $\epsilon$ tested, finding either a certificate of robustness or an adversarial example for *every* test sample. For Adv-MLP$_B$ and LP$_d$-CNN$_A$, running our verifier over the full test set takes approximately 10 hours on 8 CPUs — the same order of magnitude as the time to train each network on a single GPU. Better still, verification of individual samples is fully parallelizable — so verification time can be reduced with more computational power.

---

[7]As mentioned in Section 2, complete verifiers will obtain either a valid adversarial example or a certificate of robustness for every input given enough time. However, we do not *always* have a guarantee of robustness or a valid adversarial example for every test sample since we terminate the optimization at 1200s to provide a better picture of how our verifier performs within reasonable time limits.

[8]On SDP$_d$-MLP$_A$, the verifier in Raghunathan et al. (2018) finds certificates for 372 samples for which our verifier reaches its time limit.

Table 2: Adversarial accuracy of MNIST and CIFAR-10 classifiers to perturbations with $l_\infty$ norm-bound $\epsilon$. In every case, we improve on both 1) the lower bound on the adversarial error, found by PGD, and 2) the previous state-of-the-art (SOA) for the upper bound, generated by the following methods: [1] Kolter & Wong (2017), [2] Dvijotham et al. (2018), [3] Raghunathan et al. (2018). For classifiers marked with a ✓, we have a guarantee of robustness or a valid adversarial example for *every* test sample. Gaps between our bounds correspond to cases where the solver reached the time limit for some samples. Solve statistics on nodes explored are in Appendix F.1.

| Dataset | Network | $\epsilon$ | Test Error | Lower Bound PGD | Lower Bound Ours | Upper Bound SOA | Upper Bound Ours | No Gap? | Mean Time / s |
|---|---|---|---|---|---|---|---|---|---|
| MNIST | $LP_d$-$CNN_B$ | 0.1 | 1.19% | 2.62% | **2.73%** | 4.45%[1] | **2.74%** | | 46.33 |
| | $LP_d$-$CNN_A$ | 0.1 | 1.89% | 4.11% | **4.38%** | 5.82%[1] | **4.38%** | ✓ | 3.52 |
| | Adv-$CNN_A$ | 0.1 | 0.96% | 4.10% | **4.21%** | — | **7.21%** | | 135.74 |
| | Adv-$MLP_B$ | 0.1 | 4.02% | 9.03% | **9.74%** | 15.41%[2] | **9.74%** | ✓ | 3.69 |
| | $SDP_d$-$MLP_A$ | 0.1 | 4.18% | 11.51% | **14.36%** | 34.77%[3] | **30.81%** | | 312.43 |
| | $LP_d$-$CNN_A$ | 0.2 | 4.23% | 9.54% | **10.68%** | 17.50%[1] | **10.68%** | ✓ | 7.32 |
| | $LP_d$-$CNN_B$ | 0.3 | 11.16% | 19.70% | **24.12%** | 41.98%[1] | **24.19%** | | 98.79 |
| | $LP_d$-$CNN_A$ | 0.3 | 11.40% | 22.70% | **25.79%** | 35.03%[1] | **25.79%** | ✓ | 5.13 |
| | $LP_d$-$CNN_A$ | 0.4 | 26.13% | 39.22% | **48.98%** | 62.49%[1] | **48.98%** | ✓ | 5.07 |
| CIFAR-10 | $LP_d$-$CNN_A$ | $\frac{2}{255}$ | 39.14% | 48.23% | **49.84%** | 53.59%[1] | **50.20%** | | 22.41 |
| | $LP_d$-RES | $\frac{8}{255}$ | 72.93% | 76.52% | **77.29%** | 78.52%[1] | **77.60%** | | 15.23 |

### 5.2.1 Observations on Determinants of Verification Time

All else being equal, we might expect verification time to be correlated to the total number of ReLUs, since the solver may need to explore both possibilities for the phase of each ReLU. However, there is clearly more at play: even though $LP_d$-$CNN_A$ and Adv-$CNN_A$ have identical architectures, verification time for Adv-$CNN_A$ is two orders of magnitude higher.

Table 3: Determinants of verification time: mean verification time is 1) inversely correlated to the number of labels that can be eliminated from consideration and 2) correlated to the number of ReLUs that are not provably stable. Results are for $\epsilon = 0.1$ on MNIST; results for other networks are in Appendix F.2.

| Network | Mean Time / s | Number of Labels Eliminated | Possibly Unstable | Provably Stable Active | Provably Stable Inactive | Total |
|---|---|---|---|---|---|---|
| $LP_d$-$CNN_B$ | 46.33 | 6.87 | 311.96 | 30175.65 | 17576.39 | 48064 |
| $LP_d$-$CNN_A$ | 3.52 | 6.57 | 121.18 | 1552.52 | 3130.30 | 4804 |
| Adv-$CNN_A$ | 135.74 | 3.14 | 545.90 | 3383.30 | 874.80 | 4804 |
| Adv-$MLP_B$ | 3.69 | 4.77 | 55.21 | 87.31 | 257.48 | 400 |
| $SDP_d$-$MLP_A$ | 312.43 | 0.00 | 297.66 | 73.85 | 128.50 | 500 |

The key lies in the restricted input domain $\mathcal{G}(x)$ for each test sample $x$. When input is restricted to $\mathcal{G}(x)$, we can prove that many ReLUs are stable (with respect to $\mathcal{G}$). Furthermore, we can eliminate some labels from consideration by proving that the upper bound on the output neuron corresponding to that label is lower than the lower bound for some other output neuron. As the results in Table 3 show, a significant number of ReLUs can be proven to be stable, and a significant number of labels can be eliminated from consideration. Rather than being correlated to the *total* number of ReLUs, solve times are instead more strongly correlated to the number of ReLUs that are not provably stable, as well as the number of labels that cannot be eliminated from consideration.

## 6 DISCUSSION

This paper presents an efficient complete verifier for piecewise-linear neural networks.

While we have focused on evaluating networks on the class of perturbations they are *designed* to be robust to, *defining* a class of perturbations that generates images perceptually similar to the original remains an important direction of research. Our verifier *is* able to handle new classes of perturbations (such as convolutions applied to the original image) as long as the set of perturbed images is a union of polytopes in the input space.

We close with ideas on improving verification of neural networks. First, our improvements can be combined with other optimizations in solving MILPs. For example, Bunel et al. (2018) discusses splitting on the *input domain*, producing two sub-MILPs where the input in each sub-MILP is restricted to be from a half of the input domain. Splitting on the input domain could be particularly useful where the split selected tightens bounds sufficiently to significantly reduce the number of unstable ReLUs that need to be considered in each sub-MILP. Second, as previously discussed, taking advantage of locally stable ReLUs speeds up verification; network verifiability could be improved during training via a regularizer that increases the number of locally stable ReLUs. Finally, we observed (see Appendix H) that sparsifying weights promotes verifiability. Adopting a principled sparsification approach (for example, $l_1$ regularization during training, or pruning and retraining (Han et al., 2016)) could potentially further increase verifiability without compromising on the true adversarial accuracy.

### ACKNOWLEDGMENTS

This work was supported by Lockheed Martin Corporation under award number RPP2016-002 and the NSF Graduate Research Fellowship under grant number 1122374. We would like to thank Eric Wong, Aditi Raghunathan, Jonathan Uesato, Huan Zhang and Tsui-Wei Weng for sharing the networks verified in this paper, and Guy Katz, Nicholas Carlini and Matthias Hein for discussing their results.

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

## A   FORMULATING THE MILP MODEL

### A.1   FORMULATING ReLU IN AN MILP MODEL

We reproduce our formulation for the ReLU below.

$$y \leq x - l(1 - a) \tag{8}$$
$$y \geq x \tag{9}$$
$$y \leq u \cdot a \tag{10}$$
$$y \geq 0 \tag{11}$$
$$a \in \{0, 1\} \tag{12}$$

We consider two cases.

Recall that $a$ is the indicator variable $a = \mathbb{1}_{x \geq 0}$.

When $a = 0$, the constraints in Equation 10 and 11 are binding, and together imply that $y = 0$. The other two constraints are not binding, since Equation 9 is no stricter than Equation 11 when $x < 0$, while Equation 8 is no stricter than Equation 10 since $x - l \geq 0$. We thus have $a = 0 \implies y = 0$.

When $a = 1$, the constraints in Equation 8 and 9 are binding, and together imply that $y = x$. The other two constraints are not binding, since Equation 11 is no stricter than Equation 9 when $x > 0$, while Equation 10 is no stricter than Equation 8 since $x \leq u$. We thus have $a = 1 \implies y = x$.

This formulation for rectified linearities is sharp (Vielma, 2015) if we have no further information about $x$. This is the case since relaxing the integrality constraint on $a$ leads to $(x, y)$ being restricted to an area that is the convex hull of $y = \max(x, 0)$. However, if $x$ is an affine expression $x = w^T z + b$, the formulation is no longer sharp, and we can add more constraints using bounds we have on $z$ to improve the problem formulation.

### A.2   FORMULATING THE MAXIMUM FUNCTION IN AN MILP MODEL

We reproduce our formulation for the maximum function below.

$$y \leq x_i + (1 - a_i)(u_{max,-i} - l_i) \, \forall i \tag{13}$$
$$y \geq x_i \, \forall i \tag{14}$$
$$\sum_{i=1}^{m} a_i = 1 \tag{15}$$
$$a_i \in \{0, 1\} \tag{16}$$

Equation 15 ensures that exactly one of the $a_i$ is 1. It thus suffices to consider the value of $a_i$ for a single variable.

When $a_i = 1$, Equations 13 and 14 are binding, and together imply that $y = x_i$. We thus have $a_i = 1 \implies y = x_i$.

When $a_i = 0$, we simply need to show that the constraints involving $x_i$ are never binding regardless of the values of $x_1, x_2, \ldots, x_m$. Equation 14 is not binding since $a_i = 0$ implies $x_i$ is not the (unique) maximum value. Furthermore, we have chosen the coefficient of $a_i$ such that Equation 13 is not binding, since $x_i + u_{max,-i} - l_i \geq u_{max,-i} \geq y$. This completes our proof.

### A.3   EXPRESSING $l_p$ NORMS AS THE OBJECTIVE OF AN MIP MODEL

#### A.3.1   $l_1$

When $d(x', x) = \|x' - x\|_1$, we introduce the auxiliary variable $\delta$, which bounds the elementwise absolute value from above: $\delta_j \geq x'_j - x_j, \delta_j \geq x_j - x'_j$. The optimization in Equation 3-5 is

equivalent to

$$\min_{x'} \sum_j \delta_j \tag{17}$$

$$\text{subject to} \quad \text{argmax}_i(f_i(x')) \neq \lambda(x) \tag{18}$$

$$x' \in \mathcal{X}_{valid} \tag{19}$$

$$\delta_j \geq x'_j - x_j \tag{20}$$

$$\delta_j \geq x_j - x'_j \tag{21}$$

### A.3.2  $l_\infty$

When $d(x', x) = \|x' - x\|_\infty$, we introduce the auxiliary variable $\epsilon$, which bounds the $l_\infty$ norm from above: $\epsilon \geq x'_j - x_j, \epsilon \geq x_j - x'_j$. The optimization in Equation 3-5 is equivalent to

$$\min_{x'} \epsilon \tag{22}$$

$$\text{subject to} \quad \text{argmax}_i(f_i(x')) \neq \lambda(x) \tag{23}$$

$$x' \in \mathcal{X}_{valid} \tag{24}$$

$$\epsilon \geq x'_j - x_j \tag{25}$$

$$\epsilon \geq x_j - x'_j \tag{26}$$

### A.3.3  $l_2$

When $d(x', x) = \|x' - x\|_2$, the objective becomes quadratic, and we have to use a Mixed Integer Quadratic Program (MIQP) solver. However, no auxiliary variables are required: the optimization in Equation 3-5 is simply equivalent to

$$\min_{x'} \sum_j (x'_j - x_j)^2 \tag{27}$$

$$\text{subject to} \quad \text{argmax}_i(f_i(x')) \neq \lambda(x) \tag{28}$$

$$x' \in \mathcal{X}_{valid} \tag{29}$$

## B  DETERMINING TIGHT BOUNDS ON DECISION VARIABLES

Our framework for determining bounds on decision variables is to view the neural network as a computation graph $G$. Directed edges point from function input to output, and vertices represent variables. Source vertices in $G$ correspond to the input of the network, and sink vertices in $G$ correspond to the output of the network. The computation graph begins with defined bounds on the input variables (based on the input domain ($\mathcal{G}(x) \cap \mathcal{X}_{valid}$)), and is augmented with bounds on intermediate variables as we determine them. The computation graph is acyclic for the feed-forward networks we consider.

Since the networks we consider are piecewise-linear, any subgraph of $G$ can be expressed as an MILP, with constraints derived from 1) input-output relationships along edges and 2) bounds on the values of the source nodes in the subgraph. Integer constraints are added whenever edges describe a non-linear relationship.

We focus on computing an upper bound on some variable $v$; computing lower bounds follows a similar process. All the information required to determine the best possible bounds on $v$ is contained in the subtree of $G$ rooted at $v$, $G_v$. (Other variables that are not ancestors of $v$ in the computation graph cannot affect its value.) Maximizing the value of $v$ in the MILP $M_v$ corresponding to $G_v$ gives the optimal upper bound on $v$.

We can reduce computation time in two ways. Firstly, we can prune some edges and vertices of $G_v$. Specifically, we select a set of variables with existing bounds $V_I$ that we assume to be independent (that is, assume that they each can take on any value independent of the value of the other variables in $V_I$). We remove all in-edges to vertices in $V_I$, and eliminate variables without children, resulting in

the smaller computation graph $G_{v,V_I}$. Maximizing the value of $v$ in the MILP $M_{v,V_I}$ corresponding to $G_{v,V_I}$ gives a valid upper bound on $v$ that is optimal if the independence assumption holds.

We can also reduce computation time by relaxing some of the integer constraints in $M_v$ to obtain a MILP with fewer integer variables $M'_v$. Relaxing an integer constraint corresponds to replacing the relevant non-linear relationship with its convex relaxation. Again, the objective value returned by maximizing the value of $v$ over $M'_v$ may not be the optimal upper bound, but is guaranteed to be a valid bound.

### B.1    FULL

FULL considers the full subtree $G_v$ and does not relax any integer constraints. The upper and lower bound on $v$ is determined by maximizing and minimizing the value of $v$ in $M_v$ respectively. FULL is also used in Cheng et al. (2017) and Fischetti & Jo (2018).

If solves proceed to optimality, FULL is guaranteed to find the optimal bounds on the value of a single variable $v$. The trade-off is that, for deeper layers, using FULL can be relatively inefficient, since solve times in the worst case are exponential in the number of binary variables in $M_v$.

Nevertheless, contrary to what is asserted in Cheng et al. (2017), we *can* terminate solves early and still obtain useful bounds. For example, to determine an upper bound on $v$, we set the objective of $M_v$ to be to maximize the value of $v$. As the solve process proceeds, we obtain progressively better certified upper bounds on the maximum value of $v$. We can thus terminate the solve process and extract the best upper bound found at any time, using this upper bound as a valid (but possibly loose) bound on the value of $v$.

### B.2    LINEAR PROGRAMMING (LP)

LP considers the full subtree $G_v$ but relaxes all integer constraints. This results in the optimization problem becoming a linear program that can be solved more efficiently. LP represents a good middle ground between the optimality of FULL and the performance of IA.

### B.3    INTERVAL ARITHMETIC (IA)

IA selects $V_I$ to be the parents of $v$. In other words, bounds on $v$ are determined solely by considering the bounds on the variables in the previous layer. We note that this is simply interval arithmetic Moore et al. (2009).

Consider the example of computing bounds on the variable $\hat{z}_i = W_i z_{i-1} + b_i$, where $l_{z_{i-1}} \leq z_{i-1} \leq u_{z_{i-1}}$. We have

$$\hat{z}_i \geq W_i^- u_{z_{i-1}} + W_i^+ l_{z_{i-1}} \tag{30}$$

$$\hat{z}_i \leq W_i^+ u_{z_{i-1}} + W_i^- l_{z_{i-1}} \tag{31}$$

$$W_i^+ \triangleq \max(W_i, 0) \tag{32}$$

$$W_i^- \triangleq \min(W_i, 0) \tag{33}$$

IA is efficient (since it only involves matrix operations for our applications). However, for deeper layers, using interval arithmetic can lead to overly conservative bounds.

## C    PROGRESSIVE BOUNDS TIGHTENING

GETBOUNDSFORMAX finds the tightest bounds required for specifying the constraint $y = \max(xs)$. Using the observation in Proposition 1, we stop tightening the bounds on a variable if its maximum possible value is lower than the minimum value of some other variable. GETBOUNDSFORMAX returns a tuple containing the set of elements in $xs$ that can still take on the maximum value, as well as a dictionary of upper and lower bounds.

GETBOUNDSFORMAX($xs, fs$)

```
 1   ▷ fs are the procedures to determine bounds, sorted in increasing computational complexity.
 2   d_l = {x : −∞ for x in xs}
 3   d_u = {x : ∞ for x in xs}
 4   ▷ initialize dictionaries containing best known upper and lower bounds on xs
 5   l_max = −∞      ▷ l_max is the maximum known lower bound on any of the xs
 6   a = {xs}
 7   ▷ a is a set of active elements in xs that can still potentially take on the maximum value.
 8   for f in fs:    ▷ carrying out progressive bounds tightening
 9       do for x in xs:
10          if d_u[x] < l_max
11             then a.remove(x)    ▷ x cannot take on the maximum value
12             else  u = f(x, boundType = upper)
13                   d_u[x] = min(d_u[x], u)
14                   l = f(x, boundType = lower)
15                   d_l[x] = max(d_l[x], l)
16                   l_max = max(l_max, l)
17   return (a, d_l, d_u)
```

## D  ADDITIONAL EXPERIMENTAL DETAILS

### D.1  NETWORKS USED

The source of the weights for each of the networks we present results for in the paper are provided below.

- MNIST classifiers not designed to be robust:
  - MLP-2×[20] and MLP-3×[20] are the MNIST classifiers in Weng et al. (2018), and can be found at https://github.com/huanzhang12/CertifiedReLURobustness.
- MNIST classifiers designed for robustness to perturbations with $l_\infty$ norm-bound $\epsilon = 0.1$:
  - $LP_d$-CNN$_B$ is the *large* MNIST classifier for $\epsilon = 0.1$ in Wong et al. (2018), and can be found at https://github.com/locuslab/convex_adversarial/blob/master/models_scaled/mnist_large_0_1.pth.
  - $LP_d$-CNN$_A$ is the MNIST classifier in Kolter & Wong (2017), and can be found at https://github.com/locuslab/convex_adversarial/blob/master/models/mnist.pth.
  - Adv-CNN$_A$ was trained with adversarial examples generated by PGD. PGD attacks were carried out with $l_\infty$ norm-bound $\epsilon = 0.1$, 8 steps per sample, and a step size of 0.334. An $l_1$ regularization term was added to the objective with a weight of 0.0015625 on the first convolution layer and 0.003125 for the remaining layers.
  - Adv-MLP-2×[200] was trained with adversarial examples generated by PGD. PGD attacks were carried out with with $l_\infty$ norm-bound $\epsilon = 0.15$, 200 steps per sample, and a step size of 0.1. An $l_1$ regularization term was added to the objective with a weight of 0.003 on the first layer and 0 for the remaining layers.
  - $SDP_d$-MLP-1×[500] is the classifier in Raghunathan et al. (2018).
- MNIST classifiers designed for robustness to perturbations with $l_\infty$ norm-bound $\epsilon = 0.2, 0.3, 0.4$:
  - $LP_d$-CNN$_A$ was trained with the code available at https://github.com/locuslab/convex_adversarial at commit 4e9377f. Parameters selected were batch_size=20, starting_epsilon=0.01, epochs=200, seed=0.
  - $LP_d$-CNN$_B$ is the *large* MNIST classifier for $\epsilon = 0.3$ in Wong et al. (2018), and can be found at https://github.com/locuslab/convex_adversarial/blob/master/models_scaled/mnist_large_0_3.pth.

- CIFAR-10 classifiers designed for robustness to perturbations with $l_\infty$ norm-bound $\epsilon = \frac{2}{255}$
    - $LP_d$-CNN$_A$ is the *small* CIFAR classifier in Wong et al. (2018), courtesy of the authors.
- CIFAR-10 classifiers designed for robustness to perturbations with $l_\infty$ norm-bound $\epsilon = \frac{8}{255}$
    - $LP_d$-RES is the *resnet* CIFAR classifier in Wong et al. (2018), and can be found at `https://github.com/locuslab/convex_adversarial/blob/master/models_scaled/cifar_resnet_8px.pth`.

## D.2 COMPUTATIONAL ENVIRONMENT

We construct the MILP models in Julia (Bezanson et al., 2017) using JuMP (Dunning et al., 2017), with the model solved by the commercial solver Gurobi 7.5.2 (Gurobi Optimization, 2017). All experiments were run on a KVM virtual machine with 8 virtual CPUs running on shared hardware, with Intel(R) Xeon(R) CPU E5-2630 v4 @ 2.20GHz processors, and 8GB of RAM.

## E PERFORMANCE OF VERIFIER WITH OTHER MILP SOLVERS

To give a sense for how our verifier performs with other solvers, we ran a comparison with the Cbc (Forrest et al., 2018) and GLPK (Makhorin, 2012) solvers, two open-source MILP solvers.

Table 4: Performance of verifier with different MILP solvers on MNIST $LP_d$-CNN$_A$ network with $\epsilon = 0.1$. Verifier performance is best with Gurobi, but our verifier outperforms both the lower bound from PGD and the upper bound generated by the SOA method in Kolter & Wong (2017) when using the Cbc solver too.

| Approach | Adversarial Error | | Mean Time / s |
|---|---|---|---|
| | Lower Bound | Upper Bound | |
| Ours w/ Gurobi | 4.38% | 4.38% | 3.52 |
| Ours w/ Cbc | 4.30% | 4.82% | 18.92 |
| Ours w/ GLPK | 3.50% | 7.30% | 35.78 |
| PGD / SOA | 4.11% | 5.82% | – |

When we use GLPK as the solver, our performance is significantly worse than when using Gurobi, with the solver timing out on almost 4% of samples. While we time out on some samples with Cbc, our verifier still provides a lower bound better than PGD and an upper bound significantly better than the state-of-the-art for this network. Overall, verifier performance is affected by the underlying MILP solver used, but we are still able to improve on existing bounds using an open-source solver.

## F ADDITIONAL SOLVE STATISTICS

### F.1 NODES EXPLORED

Table 5 presents solve statistics on nodes explored to supplement the results reported in Table 2. If the solver explores zero nodes for a particular sample, it proved that the sample was robust (or found an adversarial example) without branching on any binary variables. This occurs when the bounds we find during the presolve step are sufficiently tight. We note that this occurs for over 95% of samples for $LP_d$-CNN$_A$ for $\epsilon = 0.1$.

### F.2 DETERMINANTS OF VERIFICATION TIME

Table 6 presents additional information on the determinants of verification time for networks we omit in Table 3.

Table 5: Solve statistics on nodes explored when determining adversarial accuracy of MNIST and CIFAR-10 classifiers to perturbations with $l_\infty$ norm-bound $\epsilon$. We solve a linear program for each of the *nodes explored* in the MILP search tree.

| Dataset | Network | $\epsilon$ | Mean Time /s | Nodes Explored | | | | | | |
| | | | | Mean | Median | Percentile | | | | Max |
| | | | | | | 90 | 95 | 99 | 99.9 | |
|---------|---------|-----------|---------|---------|--------|-------|-------|-------|-------|--------|
| MNIST | $LP_d$-$CNN_B$ | 0.1 | 46.33 | 8.18 | 0 | 0 | 0 | 1 | 2385 | 20784 |
| | $LP_d$-$CNN_A$ | 0.1 | 3.52 | 1.91 | 0 | 0 | 0 | 1 | 698 | 1387 |
| | Adv-$CNN_A$ | 0.1 | 135.74 | 749.55 | 0 | 201 | 3529 | 20195 | 31559 | 50360 |
| | Adv-$MLP_B$ | 0.1 | 3.69 | 87.17 | 0 | 1 | 3 | 2129 | 11625 | 103481 |
| | $SDP_d$-$MLP_A$ | 0.1 | 312.43 | 4641.33 | 39 | 17608 | 21689 | 27120 | 29770 | 29887 |
| | $LP_d$-$CNN_A$ | 0.2 | 7.32 | 15.71 | 0 | 1 | 1 | 540 | 2151 | 7105 |
| | $LP_d$-$CNN_B$ | 0.3 | 98.79 | 305.82 | 0 | 607 | 1557 | 4319 | 28064 | 185500 |
| | $LP_d$-$CNN_A$ | 0.3 | 5.13 | 31.58 | 0 | 5 | 119 | 788 | 2123 | 19650 |
| | $LP_d$-$CNN_A$ | 0.4 | 5.07 | 57.32 | 1 | 79 | 320 | 932 | 3455 | 43274 |
| CIFAR-10 | $LP_d$-$CNN_A$ | $\frac{2}{255}$ | 22.41 | 195.67 | 0 | 1 | 1 | 4166 | 29774 | 51010 |
| | $LP_d$-RES | $\frac{8}{255}$ | 15.23 | 41.38 | 0 | 1 | 3 | 1339 | 4239 | 5022 |

Table 6: Solve statistics on number of labels that can be eliminated from consideration, and number of ReLUs that are provably stable, when determining adversarial accuracy of MNIST and CIFAR-10 classifiers to perturbations with $l_\infty$ norm-bound $\epsilon$.

| Dataset | Network | $\epsilon$ | Mean Time / s | Number of Labels Eliminated | Number of ReLUs | | | |
| | | | | | Possibly Unstable | Provably Stable | | Total |
| | | | | | | Active | Inactive | |
|---------|---------|-----------|---------|---------|---------|---------|---------|---------|
| MNIST | $LP_d$-$CNN_A$ | 0.2 | 7.32 | 5.84 | 115.51 | 3202.67 | 1485.82 | 4804 |
| | $LP_d$-$CNN_B$ | 0.3 | 98.79 | 5.43 | 575.61 | 34147.99 | 13340.40 | 48064 |
| | $LP_d$-$CNN_A$ | 0.3 | 5.13 | 4.57 | 150.90 | 3991.38 | 661.72 | 4804 |
| | $LP_d$-$CNN_A$ | 0.4 | 5.07 | 2.67 | 172.63 | 4352.60 | 278.78 | 4804 |
| CIFAR-10 | $LP_d$-$CNN_A$ | $\frac{2}{255}$ | 22.41 | 7.06 | 371.36 | 4185.35 | 1687.29 | 6244 |
| | $LP_d$-RES | $\frac{8}{255}$ | 15.23 | 6.94 | 1906.15 | 96121.50 | 9468.35 | 107496 |

## G    WHICH ADVERSARIAL EXAMPLES ARE MISSED BY PGD?

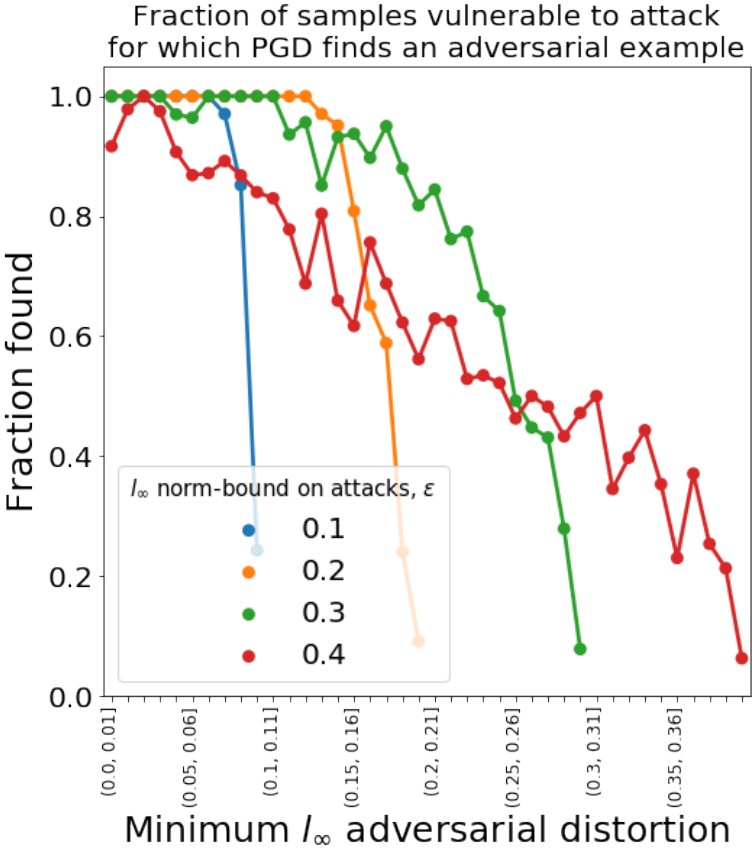

Figure 3: Fraction of samples in the MNIST test set vulnerable to attack for which PGD succeeds at finding an adversarial example. Samples are binned by their minimum adversarial distortion (as measured under the $l_\infty$ norm), with bins of size 0.01. Each of these are $\text{LP}_d\text{-CNN}_A$ networks, and were trained to optimize for robustness to attacks with $l_\infty$ norm-bound $\epsilon$. For any given network, the success rate of PGD declines as the minimum adversarial distortion increases. Comparing networks, success rates decline for networks with larger $\epsilon$ even at the same minimum adversarial distortion.

PGD succeeds in finding an adversarial example if and only if the starting point for the gradient descent is in the basin of attraction of some adversarial example. Since PGD initializes the gradient descent with a randomly chosen starting point within $\mathcal{G}(x) \cap \mathcal{X}_{valid}$, the success rate (with a single random start) corresponds to the fraction of $\mathcal{G}(x) \cap \mathcal{X}_{valid}$ that is in the basin of attraction of some adversarial example.

Intuitively, the success rate of PGD should be inversely related to the magnitude of the minimum adversarial distortion $\hat{\delta}$: if $\hat{\delta}$ is small, we expect more of $\mathcal{G}(x) \cap \mathcal{X}_{valid}$ to correspond to adversarial examples, and thus the union of the basins of attraction of the adversarial examples is likely to be larger. We investigate here whether our intuition is substantiated.

To obtain the best possible empirical estimate of the success rate of PGD for each sample, we would need to re-run PGD initialized with *multiple* different randomly chosen starting points within $\mathcal{G}(x) \cap \mathcal{X}_{valid}$.

However, since we are simply interested in the relationship between success rate and minimum adversarial distortion, we obtained a coarser estimate by binning the samples based on their minimum adversarial distortion, and then calculating the fraction of samples in each bin for which PGD with a *single* randomly chosen starting point succeeds at finding an adversarial example.

Figure 3 plots this relationship for four networks using the $\text{CNN}_\text{A}$ architecture and trained with the same training method $\text{LP}_\text{d}$ but optimized for attacks of different size. Three features are clearly discernible:

- PGD is very successful at finding adversarial examples when the magnitude of the minimum adversarial distortion, $\hat{\delta}$, is small.

- The success rate of PGD declines significantly for all networks as $\hat{\delta}$ approaches $\epsilon$.

- For a given value of $\hat{\delta}$, and two networks $a$ and $b$ trained to be robust to attacks with $l_\infty$ norm-bound $\epsilon_a$ and $\epsilon_b$ respectively (where $\epsilon_a < \epsilon_b$), PGD is consistently more successful at attacking the network trained to be robust to smaller attacks, $a$, as long as $\hat{\delta} \ll \epsilon_a$.

The sharp decline in the success rate of PGD as $\hat{\delta}$ approaches $\epsilon$ is particularly interesting, especially since it is suggests a pathway to generating networks that *appear* robust when subject to PGD attacks of bounded $l_\infty$ norm but are in fact vulnerable to such bounded attacks: we simply train the network to maximize the total number of adversarial examples with minimum adversarial distortion close to $\epsilon$.

## H    SPARSIFICATION AND VERIFIABILITY

When verifying the robustness of $\text{SDP}_\text{d}$-$\text{MLP}_\text{A}$, we observed that a significant proportion of kernel weights were close to zero. Many of these tiny weights are unlikely to be contributing significantly to the final classification of any input image. Having said that, setting these tiny weights to zero *could* potentially reduce verification time, by 1) reducing the size of the MILP formulation, and by 2) ameliorating numerical issues caused by the large range of numerical coefficients in the network (Gurobi, 2017).

We generated sparse versions of the original network to study the impact of sparseness on solve times. Our heuristic sparsification algorithm is as follows: for each fully-connected layer $i$, we set a fraction $f_i$ of the weights with smallest absolute value in the kernel to 0, and rescale the rest of the weights such that the $l_1$ norm of the kernel remains the same.[9] Note that $\text{MLP}_\text{A}$ consists of only two layers: one hidden layer (layer 1) and one output layer (layer 2).

Table 7: Effect of sparsification of $\text{SDP}_\text{d}$-$\text{MLP}_\text{A}$ on verifiability. Test error increases slightly as larger fractions of kernel weights are set to zero, but the certified upper bound on adversarial error decreases significantly as the solver reaches the time limit for fewer samples.

| Fraction zeroed | | Test Error | Certified Bounds on Adversarial Error | | Mean Time / s | Fraction Timed Out |
|---|---|---|---|---|---|---|
| $f_1$ | $f_2$ | | Lower Bound | Upper Bound | | |
| 0.0 | 0.00 | **4.18%** | 14.36% | 30.81% | 312.4 | 0.1645 |
| 0.5 | 0.25 | 4.22% | 14.60% | 25.25% | 196.0 | 0.1065 |
| 0.8 | 0.25 | 4.22% | 15.03% | **18.26%** | 69.7 | 0.0323 |
| 0.9 | 0.25 | 4.93% | 17.97% | 18.76% | 22.2 | 0.0079 |

Table 7 summarizes the results of verifying sparse versions of $\text{SDP}_\text{d}$-$\text{MLP}_\text{A}$; the first row presents results for the original network, and the subsequent rows present results when more and more of the kernel weights are set to zero.

When comparing the first and last rows, we observe an improvement in both mean time and fraction timed out by an order of magnitude. As expected, sparsifying weights increases the test error, but the impact is not significant until $f_1$ exceeds 0.8. We also find that sparsification significantly improves our upper bound on adversarial error — to a point: the upper bound on adversarial error for $f_1 = 0.9$ is higher than that for $f_1 = 0.8$, likely because the true adversarial error has increased significantly.

---

[9]Skipping the rescaling step did not appreciably affect verification times or test errors.

Starting with a network that is robust, we have demonstrated that a simple sparsification approach can already generate a sparsified network with an upper bound on adversarial error significantly lower than the best upper bound that can be determined for the original network. Adopting a more principled sparsification approach could achieve the same improvement in verifiability but without compromising on the true adversarial error as much.

# I ROBUST TRAINING AND ReLU STABILITY

Networks that are designed to be robust need to balance two competing objectives. Locally, they need to be robust to small perturbations to the input. However, they also need to retain sufficient global expressiveness to maintain a low test error.

For the networks in Table 3, even though each robust training approach estimates the worst-case error very differently, all approaches lead to a significant fraction of the ReLUs in the network being provably stable with respect to perturbations with bounded $l_\infty$ norm. In other words, for the input domain $\mathcal{G}(x)$ consisting of all bounded perturbations of the sample $x$, we can show that, for many ReLUs, the input to the unit is always positive (and thus the output is linear in the input) or always negative (and thus the output is always zero). As discussed in the main text, we believe that the need for the network to be robust to perturbations in $\mathcal{G}$ drives more ReLUs to be provably stable with respect to $\mathcal{G}$.

To better understand how networks can retain global expressiveness even as many ReLUs are provably stable with respect to perturbations with bounded $l_\infty$ norm $\epsilon$, we study how the number of ReLUs that are provably stable changes as we vary the size of $\mathcal{G}(x)$ by changing the maximum allowable $l_\infty$ norm of perturbations. The results are presented in Figure 4.

As expected, the number of ReLUs that cannot be proven to be stable increases as the maximum allowable $l_\infty$ norm of perturbations increases. More interestingly, $\text{LP}_\text{d}$-$\text{CNN}_\text{A}$ is very sensitive to the $\epsilon = 0.1$ threshold, with a sharp increase in the number of ReLUs that cannot be proven to be stable when the maximum allowable $l_\infty$ norm of perturbations increases beyond 0.102. An increase of the same abruptness is not seen for the other two networks.

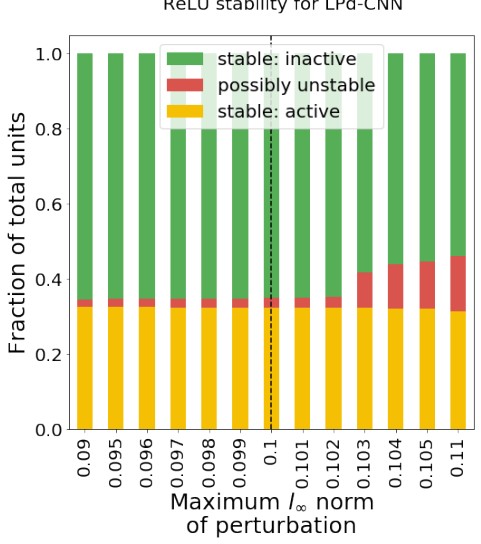

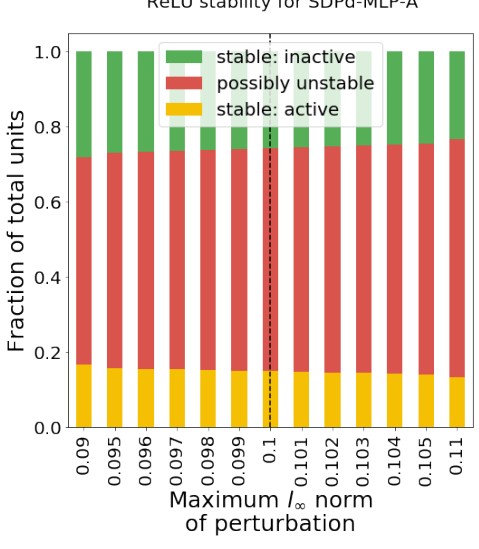

(a) $LP_d$-$CNN_A$. Note the sharp increase in the number of ReLUs that cannot be proven to be stable when the maximum $l_\infty$ norm increases beyond 0.102.

(b) $SDP_d$-$MLP_A$.

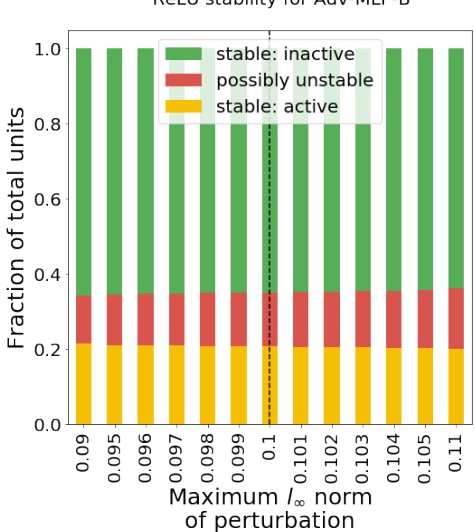

(c) Adv-$MLP_B$. Adversarial training alone is sufficient to significantly increase the number of ReLUs that are provably stable.

Figure 4: Comparison of provably ReLU stability for networks trained via different robust training procedures to be robust at $\epsilon = 0.1$, when varying the maximum allowable $l_\infty$ norm of the perturbation. The results reported in Table 3 are marked by a dotted line. As we increase the maximum allowable $l_\infty$ norm of perturbations, the number of ReLUs that cannot be proven to be stable increases across all networks (as expected), but $LP_d$-$CNN_A$ is far more sensitive to the $\epsilon = 0.1$ threshold.

