# OpenReview forum: "Evaluating Robustness of Neural Networks with Mixed Integer Programming"
_ICLR.cc/2019/Conference_

### Official Review · AnonReviewer3 · 2018-11-03
**good paper**

**Rating:** 7
**Confidence:** 1

**Review:**

This paper presents a mixed integer programming technique for verification of piecewise linear neural networks. This work uses progressive bounds tightening approach to determine bounds for inputs to units. The authors also show that this technique speeds up the bound determination by orders of magnitude as compared to other complete and incomplete verifiers. They also compare the advercerial accuracies on MNIST and CIFAR and improve on the lower bounds as compared to PGD and upper bounds as compared to SOA. The paper is well written and presents a valuable technique for evaluating robustness of classifiers to adversarial attacks.

---

> ### Author Response · Authors · 2018-11-24
> **Review Response**
>
> Thank you for your positive feedback!

---

### Official Review · AnonReviewer1 · 2018-11-03
**A strong contribution**

**Rating:** 8
**Confidence:** 5

**Review:**

This paper studies a Mixed Integer Linear Programming (MILP) approach to verifying the robustness of neural networks with ReLU activations. The main contribution of the paper is a progressive bound tightening approach that results in significantly faster MILP solving. This in turn allows for verifying the robustness of larger networks than previously studied, and even larger datasets such as CIFAR-10.

This paper is a solid contribution and should be accepted to ICLR. It is quite well-written, addresses an important problem using a principled method, and achieves strong experimental results that were previously elusive, despite the large body of work in adversarial learning. In particular, the paper has the following strengths:

- Clarity: the paper is well-written and easy to read. Tables, figures and pseudocode are nice and easy to understand.
- Methodology: the authors take care of a number of bottlenecks in the scalability of MIP solvers for the verification problem. This is the standard approach in the Operations Research (OR) community, and I am really glad to see it in an ICLR submission!
- Results: the efficiency of the MIP on the tightened model, and the improvements in the bounds on the adversarial error as compared to very recent methods from the literature are both very strong points in favor of the paper.

I do not have any further questions for the authors - good job!

---

> ### Author Response · Authors · 2018-11-24
> **Review Response**
>
> Thank you for your review. We are glad that you found our paper easy to read!
>
> Addressing the bottlenecks in the scalability of the MIP solver was key in making the verification problem tractable. We look forward to utilizing other ideas from the Operations Research community (such as computing cutting planes that exploit our knowledge of the structure of our network) to further improve performance.

---

### Official Review · AnonReviewer2 · 2018-11-04
**Strong well written paper, some improvement possible in experimental section**

**Rating:** 7
**Confidence:** 5

**Review:**

The authors perform a careful study of mixed integer linear programming approaches for verifying robustness of neural networks to adversarial perturbations. They propose three enhancements to MILP formulations of neural network verification: Asymmetric bounds, restricted domain and progressive bound tightening, which lead to significantly more scalable verification algorithms vis-a-vis prior work. They study the effectiveness of MILP solvers both in terms of verifying robustness (compared to other complete/incomplete verifiers) and generating adversarial attacks (compared to PGD attacks) and show that their approach compares favorable across a number of architectures on MNIST and CIFAR-10. They perform careful ablation studies to validate the importance of the

Quality: The paper is very well written and organized. The problem is certainly of great interest to the deep learning community, given the difficulty of properly evaluating (and then improving) defenses against adversarial attacks. The experiments are done carefully with convincing ablation studies.

Clarity: The authors explain the relevant concepts carefully and all the experimental results are clearly written and explained.

Originality: The authors propose conceptually simple but practically significant enhancements to MILP formulations of neural network verification. However, the novelty wrt https://arxiv.org/pdf/1711.00455.pdf is not discussed carefully in my view (the  asymmetric bounds were already studied in this paper, as well as a novel branch and bound strategy). The progressive bound tightening is a novel idea as far as I can see - however, the ablation experiments show that this idea is not significant in terms of performance improvement. In terms of experiments, the authors indeed obtain strong results on verified adversarial error rates and generate attacks that PGD is unable to - however, again the results do not outperform latest results (in terms of the  best achievable upper bounds on verified error rates) available well before the ICLR deadline - https://arxiv.org/pdf/1805.12514.pdf . It would be great if the authors addressed these issues in a revised version of the paper.

Significance: The work does establish a strong algorithm for complete verification of neural networks along with several ideas that are critical to obtain strong performance with this approach.

Question:
1. I am unclear on the "restricted domain" contribution claimed in the paper - is this just exploiting the fact that the inputs to the classifier are normalized to a given range, in addition to being no more than eps away from the nominal input?

Cons
1. The authors do not compare their approach to that of https://arxiv.org/pdf/1711.00455.pdf , both in terms of conceptual novelty and in terms of experimental results. In particular, it is not clear to me whether the authors' approach remains superior on domains where tight bounds on the neural networks inputs are not available, like the problems studied in the ACAS system in the ReLuPlex paper.

2. The authors' MILP solution approach relies on having access to the state of the art commercial MILP solver Gurobi. While Gurobi is free for academic research use, for large scale neural network verification applications, this does restrict use of the approach (particularly due to limited licenses being available). It would be interesting to see a comparison that uses a freely available MILP solver (like scip.zib.de) to see how critical the approach's scalability depends on the quality of the MILP solver.

3. The authors do not outperform the latest SOA numbers in terms of verified adversarial error rates on MNIST and CIFAR classifers. It would be good to see a comparison on results from https://arxiv.org/pdf/1711.00455.pdf  (I believe the training code and trained networks are available online).

---

> ### Public Comment · ~Rudy_R_Bunel1 · 2018-11-12
> **Asymmetric bounds**
>
> To add a datapoint of information with regards to the "Originality" section of the review, especially discussing the "Asymmetric bounds" contribution:
>
> I'm one of the authors of the paper that is being asked to compare to. Our first version (https://arxiv.org/pdf/1711.00455v1.pdf on Arxiv in November 2017) didn't have asymmetric bounds and we included them in a subsequent update, after reading about the idea in a previous version of the paper under review (which we cite, and highlight the difference in appendix https://arxiv.org/pdf/1711.00455v3.pdf ). Asking the authors to discuss the difference with our use of asymmetric bounds is therefore difficult, because it's their idea which we made use of.
>
> While on the subject of asymmetric bounds, would it be possible to clarify what the results of the ablation study means? When removing asymmetric bounds and instead using M = max(-l, u), could you confirm that this is only done for ReLUs that are unstable?

---

> > ### Public Comment · (anonymous) · 2018-11-13
> > **Asymmetric bounds before this paper**
> >
> > To be fair, the asymmetric bounds were first (to the best of my knowledge) used in https://arxiv.org/pdf/1705.01320.pdf. The formulation in this current paper is simply a binarized version of the same. It's a little surprising that the paper above is not cited in this context.

---

> > > ### Author Response · Authors · 2018-11-24
> > > **Asymmetric bounds**
> > >
> > > Thank you for your comment. The formulation presented in Ehlers [c] for the ReLU does correspond to our formulation when the integer constraint on a is relaxed from a∈{0,1} to 0≤a≤1. We will update our submission to reflect this, but we believe that binarizing the formulation in Ehlers to obtain our formulation is not trivial.
> > >
> > > Furthermore, viewing things from a MIP perspective can be insightful: for example, for the maximum function, relaxing the integrality constraints on the indicator variables produces a set of linear constraints complementary to those presented for the maximum function in Ehlers that is tighter when the input values x_i are closer to their upper bounds u_i.
> > >
> > > [c] Rüdiger Ehlers. "Formal Verification of Piece-Wise Linear Feed-Forward Neural Networks." https://arxiv.org/pdf/1705.01320.pdf

---

> > ### Author Response · Authors · 2018-11-24
> > **Asymmetric Bounds**
> >
> > Thank you for your comment clarifying the point on asymmetric bounds.
> >
> > To answer your question, when removing asymmetric bounds, we use M = max(-l, u) for all ReLUs, not just those that are unstable. In principle, a solver might still be able to identify all ReLUs that are stable, eliminating the associated binary variables from consideration. In practice, solves take significantly longer (mean of 0.08s vs 133.03s), and many more nodes are explored (mean of 2.05 vs. 1498.35), suggesting that not all these extraneous binary variables (added for stable ReLUs) are eliminated.

---

> ### Author Response · Authors · 2018-11-24
> **Review Response, Part I**
>
> Thank you for your review; your comments will help us in revising the paper.
>
> Comparison to Bunel et al. [a]: We consider the ideas in our paper and those in Bunel et al. to be complementary. Both our verifier and that of Bunel et al. rely on a branch-and-bound approach, and begin by solving an LP that corresponds to the MIP we formulate, but with all integrality constraints removed. In our work, branching occurs only when we split on an unstable ReLU, producing two sub-MIPs where that ReLU is fixed as active and inactive respectively. Bunel et al. observe that it is also possible to split on the _input domain_, producing two sub-MIPs where the input in each sub-MIP is restricted to be from a half of the input domain. Splitting on the input domain could be useful when tight bounds on the perturbed input are not available (as in the problems studied in the ACAS system mentioned by the reviewer), particular where the split selected tightens bounds sufficiently to significantly reduce the number of unstable ReLUs that need to be considered.
>
> We have also reached out to the authors and are working on running their verifier on the networks for which we report results in this paper, and will provide an update as soon as one is available.
>
> Solver used: We understand the reviewer's concern about having to use a commercial solver like Gurobi. While we were unable to run a comparison on the SCIP solver suggested, we were able to run a comparison on the Cbc [1] and GLPK [2] solvers, two open-source mixed integer programming solvers. Verification is run on the MNIST classifier network LPd-CNN, with ε=0.1. The results are as follows:
>
> |                  		|       Adv. Error	| Mean	|
> |                  		|  Lower	|  Upper	| Time 	|
> | Approach         	| Bound	|Bound 	| / s   	|
> |----------------------	|----------------------	|----------	|
> | Ours w/ Gurobi	|  4.38%	|  4.38%	|   3.52 	|
> | Ours w/ Cbc     	|  4.30%	|  4.82%	| 18.92 	|
> | Ours w/ GLPK  	|  3.50%	|  7.30%	| 35.78 	|
> | PGD / SOA        	|  4.11%	|  5.82%	|     --   	|
>
> When we use GLPK as the solver, our performance is significantly worse than when using Gurobi, with the solver timing out on almost 4% of samples. While we time out on some samples with Cbc, our verifier still provides a lower bound better than PGD and an upper bound significantly better than the state-of-the-art for this network. The performance of our verifier is affected by the underlying MIP solver used, but we are still able to improve on existing bounds using non-commercial solvers.
>
> We will add this table to the appendix of the paper.
>
> [1] Coin-or branch and cut (https://projects.coin-or.org/Cbc)
> [2] GNU Linear Programming Kit (https://www.gnu.org/software/glpk/). The results presented are estimates computed from 1,000 samples.
>
> [a] Rudy Bunel et al. "A Unified View of Piecewise Linear Neural Network Verification." https://arxiv.org/pdf/1711.00455.pdf

---

> ### Author Response · Authors · 2018-11-24
> **Review Response, Part II**
>
> (This is the second part of our response to the reviewer.)
>
> Comparison to Wong et al. [b]: The reviewer mentions that the results in our submission do not outperform all of the latest results in terms of upper bounds on adversarial error on MNIST and CIFAR classifiers. In particular, the reviewer was interested to see a comparison on our results with those in Wong et al. at https://arxiv.org/pdf/1805.12514.pdf [3].
>
> During the discussion period, we were able to run our verifier on all but two of the networks [4] presented in Wong et al. Results are presented below.
>
> |               	|         	|       	|        	|    Certified Bounds on Adv. Error 	|  Mean 	|
> |               	|         	|       	| Test   	|   Lower Bound	|   Upper Bound 	|  Time	|
> | Dataset	| Net	|       ε   	| Error  	|   PGD  	|  Ours 	| SOA[5]	|  Ours 	|  / s   	|
> |----------------	|----------	|----------	|----------	|----------------------	|----------	|----------	|----------	|
> | MNIST		| Small	| 0.1	|  1.21%	|  3.05%	|  3.22%	|  5.06%	|  3.22%	 |   2.55 	|
> |               	| Large	| 0.1   	|  1.19%	|  2.62%	|  2.73%	|  4.45%	|  2.74%	|  46.33	|
> |               	| Small	| 0.3   	|14.77%	|24.99%	|28.37%	|43.79%	|28.37%	|    3.71	|
> |               	| Large	| 0.3   	|11.16%	|19.70%	|24.12%	|41.98%	|24.19%	|  98.79 	|
> | CIFAR10	| Small	| 2/255 	|39.14%	|48.23%	|49.84%	|53.59%	|50.20%	|  22.41 	|
> |               	| Small	| 8/255 	|72.40%	|77.36%	|78.71%	|79.46%	|78.71%	|    0.91  	|
> |               	| Large	| 8/255 	|80.99%	|82.66%	|83.54%	|83.97%	|83.55%	|    6.01  	|
> |               	| Resnet	| 8/255 	|72.93%	|76.51%	|77.29%	|78.52%	|77.60%	|  15.23	|
>
> For all of the networks we verify, we improve upon the upper bound on adversarial error provided by the certificate in Wong et al., and also improve on the lower bound provided by PGD. We also have better overall results compared to Wong et al. over all single-model networks [6] for MNIST at ε=0.1 (2.74% vs. 3.67%), MNIST at ε=0.3 (24.19% vs. 43.10%), and CIFAR10 at ε=8/255 (77.29% vs 78.22%). We perform worse only for CIFAR10 at ε=2/255 (50.20% vs 46.11%); this is a result of us only being able to verify the `Small` network for CIFAR10 at ε=2/255, which has worse underlying robustness.
>
> Finally, in response to the reviewer's question: the "restricted domain" contribution is as described --- we use the tightest possible bounds on the perturbed input, combining the fact that the inputs to the classifier are normalized to a given range and that they are no more than ε away from the nominal input. Though simple, our results in Table 1 show that using this makes a large difference in the performance of our verifier.
>
> [3] While the paper was available before the ICLR deadline, none of the networks described (other than the `Resnet` model for CIFAR10 at ε=8/255) were available until the end of October, and we were thus unable to evaluate the performance of our verifier on these more robust networks in our initial submission. The networks are now available here: https://github.com/locuslab/convex_adversarial/tree/master/models_scaled
> [4] We were not able to verify the `Large` and `Resnet` networks for the CIFAR10 dataset at ε=2/255 due to memory issues in our implementation when determining upper and lower bounds.
> [5] We note that these SOA bounds are not the same as the robust single-model errors reported in https://arxiv.org/pdf/1805.12514.pdf, since the networks were trained with a different seed.
> [6] The full "cascade" of networks that Wong et al. present in Table 2 of their paper is not currently available for verification.
>
> [b] Eric Wong et al. "Scaling provable adversarial defenses." https://arxiv.org/pdf/1805.12514.pdf

---

### Public Comment · (anonymous) · 2018-11-23
**Scalability to > 100,000 neurons due to fewer unstable units?**

I have a question in regards to the > 100,000 neurons claim, which I hope the authors can clarify.

I tried the method from your paper on the publicly available networks from https://github.com/eth-sri/eran and observed that in many cases the MILP solver does not finish even after 24 hours on networks much smaller than the 100,000 neurons network, say a 9x200 network. This usually happens when the number of unstable ReLU units (with both + and - values)  by the presolve algorithm is  > 1000.

Indeed, as observed in your experimental section, the runtime of the MILP solver is determined by the number of unstable ReLU units and *not* the total number of ReLU units in the network.

Does it mean the approach will only work for networks where the presolve algorithm can determine that only a very small fraction of the ReLU units are unstable?  Could you please report the number of unstable units on the > 100,000 network (they are not in the paper now)?

Thanks in advance.

---

> ### Author Response · Authors · 2018-11-26
> **Scaling to larger networks and number of unstable units**
>
> Thank you for your comment.
>
> We have not encountered a network trained to be robust where many cases required more than 24 hours to solve. This is the case even for the LPd-RES CIFAR classifier, where the mean number of unstable units is 1906.15, but the mean solve time for this classifier is only 15.23s. [1]
>
> There may be two other reasons why verification takes a long time for the network you selected.
>
> Firstly, as discussed in section H in the appendix on Sparsification and Verifiability, having parameter values close to zero can lead to significantly increased verification time [2].  Two fixes are possible if this is the case. A) Without access to the training procedure, Table 7 shows that it is possible to modify the network to significantly improve verifiability at a small cost to test error, by setting some fraction of weights to zero. B) Alternatively, with access to the training procedure, adopting a principled sparsification approach could improve verifiability even further at a lower cost to test error.
>
> Secondly, attempting to verify a network not trained to be robust, such as the 9x200 network available in the repository, can lead to significantly increased verification time. We have observed that regular training (without a robustness objective) leads to networks where almost all ReLUs are unstable, even for input domains of modest size (such as an l-∞ ball of radius 0.1); in contrast, all robust training procedures we had access to produced networks where a significant fraction of ReLUs were provably stable over input domains of the same size. Fortunately, when working with a network not trained to be robust, the distance to the closest adversarial example for any input sample is typically small. When this is the case, you can reduce solve times by first attempting to find an adversarial example within a smaller input domain (such as an l-∞ ball of radius 0.01), and searching over larger input domains only when no adversarial example can be found within the smaller input domain.
>
> A quick note on stability of ReLUs: for the robust networks we verified, very few ReLUs were always provably stable for all test samples; instead, the set of possibly unstable ReLUs changes significantly between test samples.
>
> [1] We have also updated the paper so that results on the numbers of ReLUs that are provably stable are reported for all networks (either in Table 3 or in Table 6). Thank you for suggesting this!
> [2] We found that networks trained simply to minimize cross-entropy loss exhibit this behavior.

---

> > ### Public Comment · (anonymous) · 2018-11-26
> > **Benefits of the proposed method over Interval arithmetic?**
> >
> > Thank you for your response.
> >
> > It seems the method will not work well for undefended networks, but I am wondering whether there is a benefit over simple Interval analysis with defended networks?
> >
> > For undefended, in Figure 1, the 3x20 undefended net can have a maximum of 60 unstable units but your verifier seems to take longer on this than on the 100K one with 1906 unstable units. Thus, for undefended nets, the method will likely not scale for the 9x200 net I mentioned.
> >
> > For defended nets, Table 5 suggests the solver branches very rarely (not more than 3 times on 95% of the images) on the 100K net and thus even if there are 1906 branches, they are rarely explored which may explain your timings.
> >
> > During the presolve step, what percentage of nodes use only Interval arithmetic for their bounds computation? In particular, what would the results look like for the 100K net if one only uses simple Interval Arithmetic with no LP/MILP at all?
> >
> > Thanks in advance.

---

> > > ### Author Response · Authors · 2018-12-06
> > > **Verifying undefended networks, and impact of presolve step**
> > >
> > > Thank you for your comments.
> > >
> > > Verifying undefended networks: We would like to begin by clarifying that the runtimes in Figure 1 and the rest of our submission are not directly comparable. Figure 1 presents results on determining the _closest_ adversarial example; the rest of our submission presents results on finding _some_ adversarial example among perturbations with l-infinity norm bound ε (or proving that no adversarial example exists among those perturbations). Determining the closest adversarial example is always expected to take more time since it is a strictly more difficult task.
> > >
> > > To determine how well our verifier performs when determining the robustness of larger undefended networks to perturbations with bounded l-infinity norm ε, we verified the 6x100 network found here https://github.com/eth-sri/eran#experimental-results on a range of values of ε. The results are over the first 500 samples [1], and we report a timeout if solve time for a sample exceeds 120s.
> > >
> > > 		| Verified 	| Verified 	| Total
> > >          ε   	| Robust 	| Vulnerable	| Verified
> > > -----------	| ---------------	| ---------------	| -----------
> > >      0.005	|          0.966	|          0.034	|      1.000
> > >      0.010	|          0.910	|          0.046 	|      0.956
> > >      0.015	|          0.756	|          0.056	|      0.812
> > >      0.020	|          0.466	|          0.072	|      0.538
> > >      0.025	|          0.238	|          0.082	|      0.320
> > >      0.030	|          0.080	|          0.098	|      0.178
> > >
> > > These results are comparable to or better than the best results reported for this network (via DeepPoly).
> > >
> > > Presolve step: During the presolve step, a significant fraction of nodes only require interval arithmetic to compute bounds. (This is precisely what allows progressive bounds tightening to reduce the overall time spent computing bounds).
> > >
> > > In terms of the impact of using LP to compute bounds, we find that it does not significantly affect _median_ solve times; however, it _does_ make a big impact for samples that are complex and take a long time to verify. For example, for the 100K network [2], 108 (out of 10000) samples take more than 120s when using only interval arithmetic to compute bounds. Of these samples, 33 can be resolved within 120s when using LP to compute bounds.
> > >
> > > [1] The standard error rate on the first 500 samples was 2.8%.
> > > [2] We refer to this network in our submission as the CIFAR-Resnet network.

---

> > > > ### Public Comment · (anonymous) · 2018-12-06
> > > > **Use case still unclear**
> > > >
> > > > Thanks for the explanation and the additional experimental data.
> > > >
> > > > It seems for the 6x100 undefended networks, the method does not really improve over state of the art, which on top of that is an incomplete verifier (the main benefit of complete verifiers is precision gain for smaller networks). The approach is also slow, e.g., it times out for around 46% of cases for eps=0.02. It appears it would timeout even more for the 9x200 network and may prove less than the incomplete verifier. For the 100K net, the standard Interval analysis seems sufficient for 99% of cases.
> > > >
> > > > Overall, I do like the direction, but the authors need to show a use case which clearly surpasses prior work on networks beyond 3x20.

---

### Author Response · Authors · 2018-11-26
**Submission Revised**

Thank you for your comments and suggestions! We have updated the submission with revisions based on them.

---

### Meta-Review · Area_Chair1 · 2018-12-17
**Important problem, solid contribution**

**Confidence:** 4
**Recommendation:** Accept (Poster)

**Metareview:**


The paper investigates mixed-integer linear programming methods for neural net robustness verification in presence of adversarial attckas. The paper addresses and important problem, is well-written, presents a novel approach and demonstrates empirical improvements; all reviewers agree that this is a solid contribution to the field.